# Farmers' Willingness to Pay for Index-Based Livestock Insurance in the North West of South Africa

**Oluwaseun Samuel Oduniyi** *, **Michael Akwasi Antwi and Sibongile Sylvia Tekana**

School of Agriculture and Life Science, Department of Agriculture and Animal Health, College of Agriculture and Environmental Science, University of South Africa, Florida Campus, Roodepoort 1710, South Africa; antwima@unisa.ac.za (M.A.A.); tekanss@unisa.ac.za (S.S.T.)

* Correspondence: eodunios@unisa.ac.za

**Abstract:** Rural livelihoods in most developing countries are threatened by climate-related risks such as drought, flood, heat waves, storms, and so on. Although farmers have adopted several adaptation strategies, they have proven less effective than hoped. Hence, index-based livestock insurance, an innovation that significantly assists farmers to acclimatise to climate-related risks, has been proposed; and its adaptability has attracted a notable increase in other African countries. However, the success of its adoption is dependent on the inclination of the farmers to pay for the service. Accordingly, this study investigates their willingness to pay for index-based livestock insurance and its determinants, and the factors influencing the total livestock units to be insured in the North West province of South Africa. Cross-sectional data were obtained from 277 cattle farmers, drawn randomly from the study area. The contingent valuation method was applied to determine the farmers' willingness to pay; and only 10.8% were willing to pay. Simultaneously, the Heckit sample selection model was used to analyse the data to identify the factors responsible for farmers' willingness to pay and total livestock units to insure. The findings revealed that farmer's experience, age, education, marital status, awareness of insurance and household dependents were statistically significant, and influenced the maximum price R600 ($42, max willingness to pay, WTP) of those who accepted index-based livestock insurance. However, by implication, the study concluded that to adopt index-based livestock insurance in the study area among the livestock farmers, there should be policies to cater for the aforementioned factors.

**Keywords:** Climate change; Agricultural risk; Adaptive capacity; Index-based livestock insurance; WTP; CVM; Heckman selection Model (two-step); South Africa

---

## 1. Introduction

Climate change and variables such as drought, extreme heat waves and the like pose a significant threat to agricultural development, including livestock production, which has been recognised to occupy a central position in food security in South Africa. This sector is the cardinal cohesive source of support and stability for the socioeconomic state of the country; hence, it is a pertinent channel for the subsistence and maintenance of most non-metropolitan towns and rural communities, which consequently accounts for 27% of the consumer's food on a weight basis [1]. However, harsh and unpredictable weather conditions have led to a significant loss of livestock in the North West, and a similar situation was reported in the Limpopo province, with different levels of shocks such as drought and high level of livestock disease [2]. This situation has consequently worsened the already precarious food security situation in the region, an experience which has also been noted in most developing countries, since climate change is a recognised global phenomenon; and developing countries are obviously more susceptible to the damaging effects due to poor adaptive capacity [3].

Subsequently, in order to reduce household shocks arising from the inherently inconsistent nature of climate, several adaptation strategies have been adopted by the livestock farmers in the North West province of South Africa. Nonetheless, despite the adaptive scheme utilised by the rural farmers, their farming input and output are still greatly influenced by the contrariness resulting from climate alteration, since very few of them have taken on any insurance backing. However, the adoption of insurance as a sui generis plan for the reduction of the damage that necessarily follows from climate change is interlaced with problems, such as rigorous indemnity payment procedures, which subsequently results in production failure and high administrative cost, and this is because insurers do not indemnify claims promptly. Ergo, these factors have discouraged reliance on this insurance option. Hence, it is observed to be relatively unproductive in the study area and has remained underdeveloped, with a long history of ineffectiveness in most poor, rural regions in developing countries [4]. Insurance plays a vital tool as a risk management instrument to minimize the adverse effects produced by climate change. Insurance forms a financial mitigation tool which the farmers can use in the face of climate change and related events [5]. This covers for the damage caused by climate change. Similarly, risk management has been of a great concern, especially in agriculture, as food production and security need to be improved. As a result of this several studies have strained to extend farmers' beliefs and concerns about climate change [6].

To this end, an improved insurance option, known as index-based livestock insurance (IBLI), has attracted significant attention because of its functional potentiality that enables farmers to firmly adapt to the inevitability of climate change, unlike conventional or traditional insurance. It is an insurance method introduced by the International Livestock Research Centre (ILRI), and regarded as one of the modern risk-management tools that will enable farmers to adroitly manage the negative shock that necessarily accompanies climate change. As the denotation indicates, this insurance scheme aims to disburse payments in accordance with the index of aggregated criteria, such as livestock losses over a geographic area, rather than households' or businesses' actual, individual losses. Subsequently, in consideration of the aforesaid, regarding the intricacies associated with climate change, agriculture and food security (CCAFS), index-based livestock insurance (IBLI) products are not only a representation of a hopeful and creative innovatory design but also an advantageous conception that will allow for easy and quick disbursement of benefits from the insurer, leading to the needed protection from the climate-related risks that vulnerable rural smallholder farmers and livestock keepers experience. Even though index-based insurance is a relatively new product, it has been implemented in developing countries, especially in Africa.

Besides, previous studies affirm that this concept has gained widespread interest over the last decade as a viable implement for the reduction of uninsured covariate risk in poor rural areas that naturally have no means of receiving commercial insurance products [7]. The essence of inaugurating this distinct form of insurance is to inspire rural development that will subsequently give room for better acclimatisation to climate change by smallholder farmers [8]. Moreover, index-based insurance gives meaningful potentiality capable of bettering agriculture beyond the provision offered by traditional insurance in such a way that indemnity payments are not reliant on individual claims. In contrast, the insurance companies and insured clients are only expected to keep tabs on the index so as to be aware of the time when payments are due. Due to this, there is a remarkable and distinct reduction in the transaction costs of observing and confirming losses, besides getting rid of some notable structural problems that plague conventional insurance, including moral hazard, untoward selection, and prevalent risks [9].

Another advantage is that index-based insurance indemnity payouts are determined based on well-defined and easily discernible objective weather or environmental parameters, such as rainfall, temperature, or remotely sensed estimates of vegetation levels that are highly correlated with losses, instead of actual losses undergone by policyholders [10,11]. Consequently, this insurance scheme makes it impossible for an individual to misrepresent or falsify the record. Essentially, it affords the insurers the opportunity to forefend, not only the hazard associated with moral standard, but also the

adverse selection problems connected to the indemnity of losses specific to the insured. The scheme also enables a remarkable reduction in transaction costs that the insurer would have expended on close observation of the behaviour of the insured and its validation [12]. Research conducted by [13] evidenced the utilisation of simulations on household-level performance analysis among East African pastoralists, which consequently evinces that IBLI eliminates 25–40 percent of the overall livestock mortality risk.

Basically, index-based insurance provides relatively transparent contracts, thereby making it possible for insurance companies to transfer their risk to international reinsurance markets [13]. Equally, it offers opportunities for households to avoid associated problems regarding the indemnity of losses that are demonstrably related to the insured in traditional insurance. Nevertheless, despite the provably and concrete evidence that shows that farmers are obviously affected by natural risks, South Africa is yet to introduce IBLI. Neither is there any manifestly empirical evidence provided on the acceptability of index-based livestock insurance by local farmers, who happen to be the key stakeholders in such interventions. The current study aspires to fill the gap by helping the farmers to improve on climate change adaptation strategies through the purchase of index-based livestock insurance, as opposed to the traditional insurance scheme. The study is different in such a way that there is no or little literature on livestock farmers' willingness to pay for IBLI in South Africa. The IBLI is a new program which this study intends to explore for the purpose of future introduction. The study is unique as it contributes to the body of knowledge on WTP for IBLI by initiating a blueprint for how much a farmer is willing to pay for IBLI and how many total livestock units (TLUs) to be insured.

Given this, the study examines the determinants and smallholder farmers' willingness to pay for index-based livestock insurance, in addition to identifying and explicating factors influencing the TLUs to be insured. Significantly, this study will create awareness by contributing to the growing literature on the uptake of index-based livestock insurance in South Africa, and categorising the vital challenges that will effectuate a notable rescale and augmentation of this insurance plan as a climate change adaptation strategy.

## 2. Study Area

**Research area**: The study was carried out in the North West province of South Africa. It is a country located at the southern tip of Africa, with a land area of 1,233,404 km$^2$ and adjoined on three sides by nearly 3,000 km of coastline, with the Indian Ocean to the east and the Atlantic Ocean to the west. There is a borderline in the north between the country and Namibia, Botswana, Zimbabwe, and Mozambique, in addition to enclosing two independent countries, the kingdoms of Lesotho and Swaziland. The North West is situated towards the western part of South Africa. It adjoins Limpopo to the North, Gauteng to the east, the Free State to the east and south, the Northern Cape to the south and Botswana to the west and north. Altitude ranges from 1,000 to 2,000 m above sea level. The total land area of the province is 116,320 km$^2$, and it occupies 9.5% of the total area of South Africa.

The province is largely rural and the major line of work is agriculture, although it is just the fifth main contributor to the GDP. The North West is segmented into four district municipalities. The two district municipalities earmarked for the study were Bojanala and Ngaka Modiri Molema (NMM), which is the capital of the province, and situated at the centre of the province as shown in Figure 1.

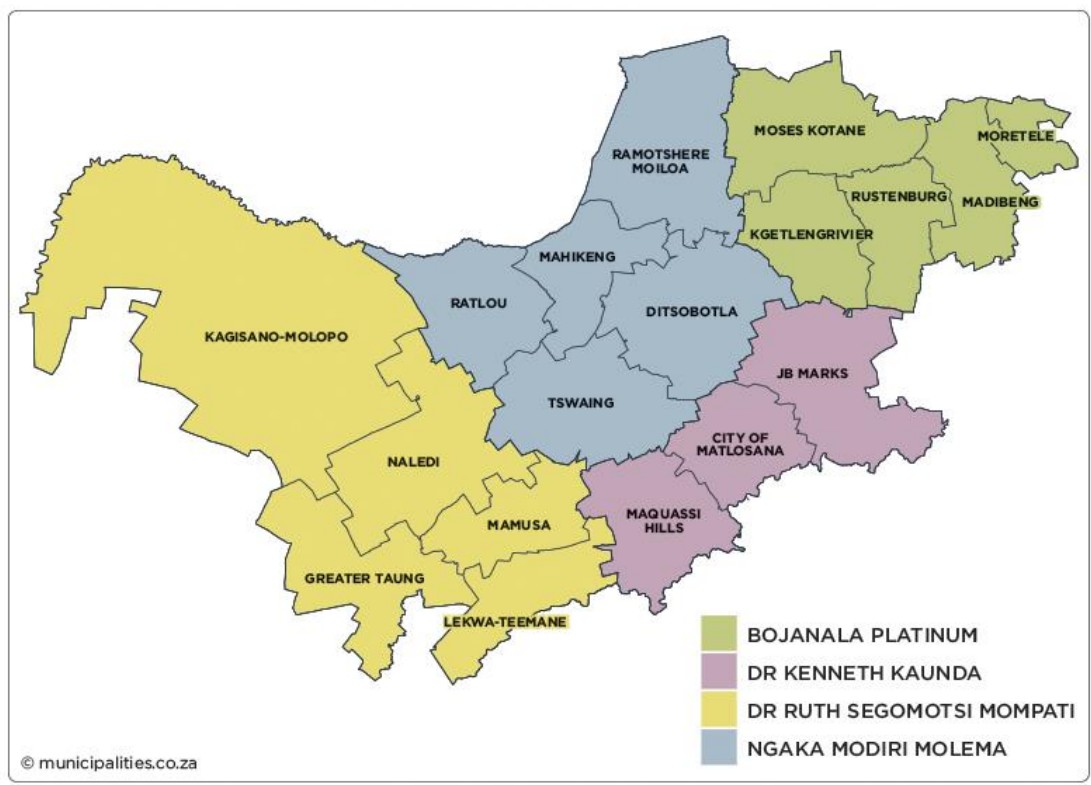

**Figure 1.** Map of North West municipalities, showing the study area. Source: https://municipalities.co.
za/provinces/view/8/north-west.

## 3. Methodology

**Population, sampling procedure and sample size:** A list of registered cattle farmers was obtained from the Provincial Department of Agriculture and Rural Development, which formed the population of 980. A representative sample size was determined using Slovin's formula, as given in Equation (1), and afterwards a total of 277 questionnaires were doled out to the cattle farmers in the districts using a random sampling technique. In accordance with [14] postulations, this was realised by adopting a quantitative model as illustrated below:

$$n = \frac{N}{1 + N(e)2} \tag{1}$$

Where n is the sample size,
N = total population of 780 cattle farmers
e = maximum variability or margin of error (MoE). This is estimated at 5% (0.05),
1 = probability of the event occurring,
277 = the number of respondents sampled or sample size.
Table 1 shows the distributions of sample size across the two districts.

**Table 1.** Sample size.

| District Municipality | Sample Size |
|---|---|
| Bojanala | 52 cattle farmers |
| Ngaka Modiri Molema | 225 cattle farmers |
| Total | 277 cattle farmers |

Source: Author's computation.

**Method of data collection:** Data were collected from the cattle farmers through an interview schedule using a semi-structured questionnaire, which was validated by two experts in the department of agricultural economics. The questionnaire was subdivided into sections based on the objectives of the study. A reliability test was done on the research instrument. The Pearson product-moment correlation method and split-half technique were employed in order to ascertain the regularity of the instrument.

**Method of data analytical techniques:** Both descriptive and inferential statistics were used for the analytic method. Descriptive statistics such as frequency counts, percentage, mean values, and standard deviation interpreted farmers' socioeconomic and farm-based characteristics. The contingent valuation method and the willingness-to-pay approach were used to analyse the index-based livestock insurance, while inferential statistics such as the Heckman sample selection model (two-steps model) were used to examine the determinants and smallholder farmers' willingness to pay for index-based livestock insurance, as well as factors influencing the total livestock units (TLUs) to be insured.

**Design of Index-Based Livestock Insurance (IBLI):** Index-based insurance, which is also known as index insurance, is primarily used in agriculture. The insurance product covers drought-related livestock mortality. This method of insurance is designed to protect farmers from every climate related threat, but only when there is a prevalent risk that expressly influences farmers livelihood. In index insurance, the indemnity is not paid to the individual farmers, but is based on an index that is related to the underlying risk a farmer is willing to cover, which is based as well as on the geographical average of the area of interest. The pay-outs are related to an index which is associated with losses in agricultural production, such as rainfall, drought, reduced pasture for livestock, and the same pay-oust are paid when the index exceeds the threshold (trigger). However, the study area has experienced severe drought since 2008, which may explain the high incidence of drought-related deaths among adult animals, as well as high mortality among small herds. Drought contributes about 60% of livestock mortality. In 2016, [15], reported that most deaths in the study area were caused by diseases (50%), followed by drought (34%). According to [16]. the index is designed through the use of satellite-based readings of forage availability in correspondence to livestock mortality data. The data is subjected to a remote normalized difference vegetation index (NDVI). The NDVI is a remote sensor which is a satellite of the US National Oceanic and Atmospheric Agency. The concept of index-based insurance was explained to the farmers in order for them to understand, and possibly be willing to pay for, index insurance, and to determine how many livestock units they are keen to insure.

**Contingent valuation method (CVM):** The contingent valuation method (CVM) is a technique that is employed for approximating the value placed on goods and services by an individual. The contingent valuation technique is an appraisal technique, and functions as an elemental economic tool for the estimation of the values of non-marketed goods. It is a direct approach used to measure the willingness to pay (WTP) in obtaining specific goods and services, or willingness to accept (WTA) to give up goods. This method has been used widely and proven useful, either singly or in conjunction with other evaluation methods for non-market goods. For instance, it has been used to evaluate environment and health care programmes [17]. Likewise, many researchers who have adopted CVM have been able to give a tangible result and predict the number of non-market goods, such as connections to water supply systems, improved conditions, and revenue for the local water authority [18]. With respect to this study, and in accordance with the objectives of this study, the willingness-to-pay method of CVM was used to determine the amount an individual is willing to pay for IBLI.

**Willingness to pay (WTP):** The willingness-to-pay approach poses questions by employing dichotomous choice techniques, that is, enquiring from an individual whether or not he/she would readily buy the specified commodity at the stipulated price. Willingness to pay for a product or service is expressed as the amount of money that a person is willing to pay for buying a product or service

from his/her income with risk preferences, while simultaneously, keeping his/her utility constant. This can be expressed mathematically:

$$V(y-WTP, p, q_1; Z) = V(y, p, q_0; Z) \tag{2}$$

where V represents the indirect utility function, y signifies income, p is a signification for the vector of prices faced by the individual, and q0 and q1 are connotations for the alternative levels of the good or quality indexes (with q1 > q0, signifying that q1 refers to better environmental quality).

Furthermore, WTP values are calculated as follows:

$$\varepsilon = WTP = \sum_{i=1}^{n} \alpha_i Pr_i \tag{3}$$

The WTP could be a single-bounded model or double-bounded model. For the purpose of this research, a single-bounded model with the open-ended contingent valuation technique was applied to accommodate the true amount a farmer would be willing to pay for index-based livestock insurance. This was determined by asking the individual farmer to put the amount they are willing to pay for IBLI in the space provided in the questionnaire. The results are shown in Table 2. This technique has been used in many studies, including those by [7,19–21]. This approach is preferred because it gives a precise and objective interpretation, hence, it is easy to know the exact price an individual is willing to pay—the 'take-it-or-leave-it approach'—and an individual would not be restricted by defined values. Lastly, this technique enables a researcher to estimate summary descriptive statistics, such as the mean, minimum and maximum price, for the sake of policymaking.

**Table 2.** Amount willing to pay for IBLI.

| Amount Willing to Pay for IBLI | Frequency | Percentage |
|:---:|:---:|:---:|
| 0 | 247 | 89.2 |
| 400 | 3 | 1.1 |
| 450 | 7 | 2.5 |
| 500 | 11 | 4.0 |
| 550 | 1 | 4 |
| 600 | 8 | 2.9 |
| Total | 277 | 100.0 |

Source: Author's computation, 2019.

Though the double-bounded model depicts more efficient welfare measures because it requires that the respondent provide more information [22], the model is considered disputable as a result of possible contradictoriness in the answers to the first binary questions, since it initiates a larger ambit for biased results [23]. However, according to [24], there are no relevant differences estimated by the two models.

**Heckman sample selection method:** The Heckman sample selection model simultaneously investigates factors that influence willingness to pay and the total livestock units to be insured in a single model. The model can explicitly solve the potential of sample selection bias, and it involves two stages, which are the selection stage and the outcome stage. While the selection stage represents a probit regression model, in contrast, the outcome model is expressed using ordinary linear regression. The Heckman sample selection model can be mathematically presented as follows:

$$\text{Stage 1: } d_i^* = \alpha x_{1i} + e \qquad \text{(outcome equation)} \tag{4}$$

$$d_i = 1 \qquad \text{if } d_i^* \geq 0$$

$$d_i = 0 \qquad \text{if } d_i^* < 0$$

$$\text{stage 2: } y_i^* = \beta x_{2i} + u \qquad \text{(selection equation)} \tag{5}$$

$$y_i = y_i^* \qquad \text{if } d_i = 1$$

$$y_i \text{ is missing} \qquad \text{if } d_i = 0$$

where:

$d_i$ and $y_i$ are the observed realisations,

$d_i^*$ and $y_i^*$ are their latent counterparts,

$x_1$ and $x_2$ are vectors of exogenous variables,

$\alpha$ and $\beta$ are unknown parameter vectors,

e and u are the corresponding error terms.

The above equation can be expressed as:

$$Y_i^* = X_{1i} B_1 + u_i \tag{6}$$

$$Z_i^* = X_{2i} B_2 + 1_i \ (i = 1, 2 \dots , I) \tag{7}$$

Equation (4) is the outcome equation and Equation (5) is the selection equation. Where $X_{ji}$ is a $1 \times K_j$ vector of exogenous regressors: $B_j$ is a $K_j \times 1$ vector of parameters; $u_i$ and $1_i$ are the error terms of outcome and selection equations, respectively. The Heckman model assumes that the error terms $u_i$ and $1_i$ a are independently and jointly normally distributed. Heckman's two-step model is preferred for this study because the estimators from this procedure are consistent and asymptotically normal, and moreover, it is more robust than the Heckman MLE approach [25]. The estimation steps are as follows: in the first step, the probit regression is used to model the sample selection process in Equation (5), and then the inverse Mills ratio $\lambda$, the error from the probit equation explicating selection, is calculated based on the probit regression results; in the second step, the inverse Mills ratio is added to multiple regression analysis as an independent variable, and the ordinary least square is used to provide the consistent parameter estimate. Table 3 shows the variables used in the model.

**Table 3.** Description of variables used in the Heckman sample model.

| Explanatory Variables | | |
| --- | --- | --- |
| **Variables** | **Description and Unit of Measurement** | **Expected Sign** |
| WTP for IBLI | Binary, 1 is the willingness to pay for IBLI and 0 if otherwise | + |
| Farming experience | Categorical, numbers of years of farming | + |
| Age | Continuous, the age of the respondent in years | + |
| Education | Binary, 1 if a farmer has formal education and 0 if otherwise | + |
| Household dependents | Continuous, number of households | − |
| Marital status | Binary, 1 if a farmer is married and 0 if otherwise | + |
| Access to credit | Binary, 1 if a farmer has access and 0 if no | + |
| Awareness of insurance | Binary, 1 if aware of insurance and 0 if otherwise | + |

Source: Author's Computation, 2019.

## 4. Results and Discussion

Tables 4 and 5 show the association of variables and the willingness to pay for IBLI. The tables also reveal descriptive statistics, such as the mean, mean difference, standard error, standard error difference and standard deviation, as well as the proportion of dummy variables influencing the willingness to pay or not to pay for IBLI. The independent t-test results in Table 4 evinces that there is a significant mean difference between farmers who are willing to pay for IBLI and those who are not with regards to farming experience, household dependents, and income. The chi-square analysis in Table 5 presents the proportion of farmers who are willing to pay for IBLI and those who are not, with respect to their socioeconomic characteristics such as gender, marital status, education and other sources of income. Tables 2 and 6 articulate the descriptive statistics and the maximum total livestock units a farmer is willing to insure in the IBLI programme at a maximum price of R600. Table 7 reveals that only 10.8% of the livestock farmers are willing to pay for IBLI at a maximum price of R600 per unit of livestock. Figure 2, shows the graph which explains willingness to pay for IBLI, the same as Table 7.

**Table 4.** Association between continuous variables and the willingness to pay (WTP) for IBLI.

| Variables | Pooled Mean | Pooled Standard Error | Pooled Standard Deviation | WTP Mean, Standard Error, Standard Deviation | Non-WTP Mean, Standard Error, Standard Deviation | Mean Difference | Standard Error Difference | *t*-Value |
|---|---|---|---|---|---|---|---|---|
| Farming experience | 9.613 | 1.247 | 6.828 | (14.267) (1.247) (6.826) | (9.049) (0.314) (4.938) | −5.218 | 1.000 | −5.221 *** |
| Age | 50.47292 | 0.476 | 7.923 | (51.5) (1.394) (7.634) | (50.348) (0.507) (7.964) | −1.152 | 1.533 | −0.751 |
| Household dependents | 6.007 | 0.172 | 2.868 | (7.133) (0.579) (3.170) | (5.870) (0.178) (2.805) | −1.263 | 0.550 | −2.295 *** |
| TLUs | 69.415 | 6.695 | 111.420 | (56.400) (5.577) (30.547) | (70.996) (7.473) (117.4529) | 14.596 | 21.564 | 0.677 |
| Income | 19303.970 | 1563.355 | 26019.410 | (33000) (1054.819) (5777.483) | (17640.49) (1719.126) (27018.19) | −15359.510 | 4953.996 | −3.100 *** |

Source: Author's computation, 2019. ***, **, * Significant at 1, 5, and 10 percent significance level, respectively.

**Table 5.** Association between dummy variables and the WTP for IBLI.

| Variables | WTP (%) | Non-WTP (%) | Pearson's Chi-Square | Likelihood Ratio | Linear-by-Linear Association |
|---|---|---|---|---|---|
| Gender | | | 3.485[a] *** | 2.357 | 6.278 |
| Male | 30 (100) | 221 (89.47) | | | |
| Female | 0 (0) | 26 (10.53) | | | |
| Marital status | | | 23.294[a] *** | 21.423 | 23.310 |
| Married | 6 (20) | 162 (65.89) | | | |
| Unmarried | 24 (80) | 85 (34.41) | | | |
| Education | | | 35.538[a] *** | 33.222 | 37.331 |
| Literate | 3 (10) | 164 (66.40) | | | |
| Non-literate | 27 (90) | 83 (33.60) | | | |
| Access to credit | | | 1.590[a] | 0.107 | 1.102 |
| Yes | 1 (3.3) | 2 (0.8) | | | |
| No | 29 (96.7) | 245 (99.2) | | | |
| Member of organisation | | | 0.368[a] | 0.000 | 0.692 |
| Yes | 0 (0) | 3 (1.2) | | | |
| No | 30 (100) | 244 (98.8) | | | |
| Other income sources | | | 2.190[a] ** | 1.647 | 2.151 |
| Yes | 14 (43.6) | 150 (60.7) | | | |
| No | 16 (53.3) | 97 (39.3) | | | |
| Access to extension visit | | | 1.130[a] | 0.268 | 2.100 |
| Yes | 0 (0) | 9 (3.6) | | | |
| No | 30 (100) | 238 (96.4) | | | |
| Landowner | | | 1.260[a] | 0.365 | 2.338 |
| Yes | 0 (0) | 10 (4.0) | | | |
| No | 30 (100) | 237 (96.0) | | | |

Source: Author's computation, 2019. ***, **, * Significant at 1, 5, and 10 percent significance level, respectively.

**Table 6.** Descriptive statistics.

| Variable | Mean | Std. Dev. | Min | Max |
|---|---|---|---|---|
| WTP for IBLI | 0.108 | 0.311 | 0 | 1 |
| TLUs willing to insure | 2.82 | 9.001 | 0 | 50 |
| Amount WTP for IBLI | 54.87 | 159.209 | 0 | 600 |
| Farming experience | 9.614 | 5.410 | 0 | 23 |
| Age | 50.472 | 7.923 | 30 | 69 |
| Education | 0.603 | 0.490 | 0 | 1 |
| Household dependents | 6.007 | 2.868 | 0 | 12 |
| Marital status | 0.606 | 0.489 | 0 | 1 |
| Access to credit | 0.011 | 0.104 | 0 | 1 |
| Awareness of insurance | 0.307 | 0.462 | 0 | 1 |

Source: Author's computation, 2019.

**Table 7.** Willingness to pay for index-based insurance.

| Willingness to pay for IBLI | Frequency | Percentage (%) | Cumulative percentage (%) |
|---|---|---|---|
| Yes | 30 | 10.8 | 10.8 |
| No | 247 | 89.2 | 100.0 |

Note: The maximum amount that a respondent is willing to pay for a TLU of cattle is R600. Source: Author's computation, 2019.

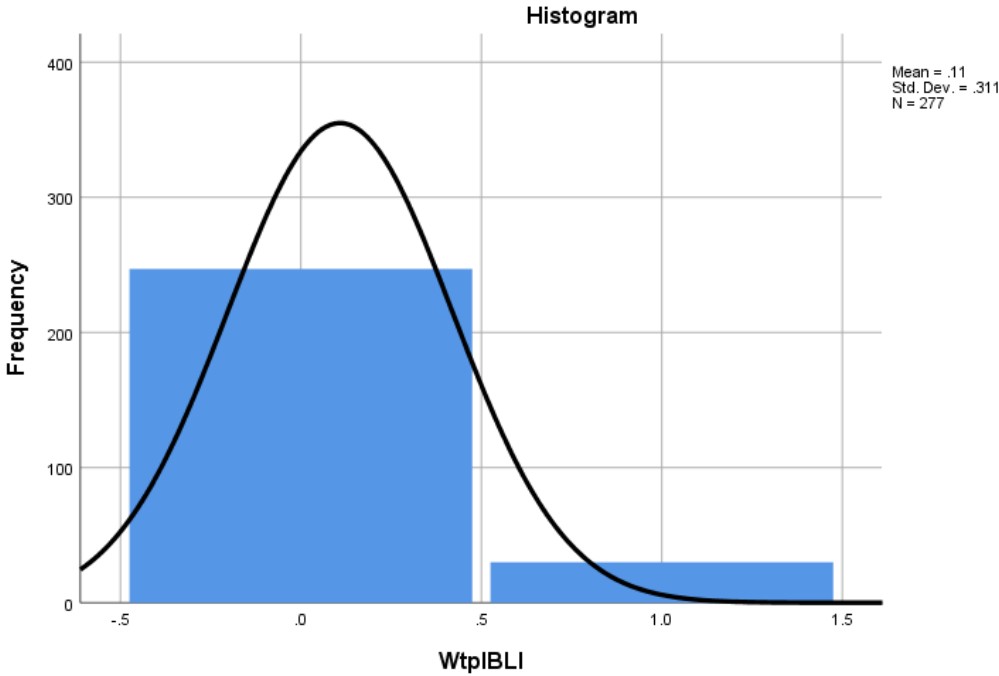

**Figure 2.** Graph showing the willingness to pay for index-based livestock insurance (IBLI).

Furthermore, the farmers' WTP for IBLI and the amount of TLUs to insure are estimated using Heckman's two-step model and employing Stata software, version 15.0. The results in Table 8 consist of two stages. The selection stage shows that farming experience, age, education, marital status and awareness of insurance are statistically significant and influences the WTP for IBLI. The farming experience of the farmers is positively significant and determines the WTP for IBLI. This can be attributed to the fact that the experienced farmers have more understanding and increased perception of climate change variabilities and their impact on livestock production, which prompts the farmer to be willing to pay for IBLI. Similarly, awareness of insurance is positively and statistically significant. The examination evidenced that farmers who are aware of insurance find it easy and comfortable to pay for IBLI. This result aligns with the research carried out by [7]. The researchers aver that a household that is aware of the programme, besides possessing the fundamental knowledge regarding the program's benefits, has a higher likelihood of accepting and paying for IBLI than the one that lacks even the rudiments and basics of the scheme.

In addition, a negative correlation was found between the age of the farmers and the willingness to pay for IBLI, although age is statistically significant but negatively correlated. It is probable that this outcome is due to the fact that older farmers are hesitant and slow to adopt a new programme, which results in their unwillingness to pay for IBLI. This was also confirmed by [26], who found that age influenced willingness to pay for IBLI. Regarding the education status of the farmers, the analysis shows that it is negatively correlated and statistically significant. The more educated the farmer is, the lesser the willingness to pay for IBLI. Ideally, farmers with better access to education are more likely to pay for indexed-based livestock insurance; but this is in contrast with the findings in the study area. The contrast may be attributed to the fact that IBLI is a relatively new programme in South Africa, and farmers have no information or are unaware of the programme; thus, the educated farmers are skeptical, and consequently not really willing to pay for IBLI. However, [27] explained that farmers who are literate are more interested in paying for insurance.

**Table 8.** Heckman selection model—two-step estimates. (regression model with sample selection).

| Variables | Coef. | Std. Err. | z | P>\|z\| |
|---|---|---|---|---|
| **WTP for IBLI** | | | | |
| Farming experience | 0.015 | 0.007 | 2.27** | 0.023 |
| Age | −0.010 | 0.006 | −1.75** | 0.080 |
| Education | −0.243 | 0.084 | −2.90*** | 0.004 |
| Household dependents | 0.008 | 0.012 | 0.66 | 0.509 |
| Marital status | −0.128 | 0.069 | −1.84** | 0.065 |
| Access to credit | 0.229 | 0.292 | 0.78 | 0.433 |
| Awareness of insurance | 0.176 | 0.067 | 2.62*** | 0.009 |
| constant | 0.558 | 0.344 | 1.85 | 0.070 |
| **TLUs willing to insure** | | | | |
| Farming experience | −0.018 | 0.055 | −0.34 | 0.736 |
| Household dependent | −0.508 | 0.280 | −1.81** | 0.070 |
| Own the Land | −0.059 | 0.441 | −0.130 | 0.894 |
| constant | 7.059 | 2.509 | 2.89 | 0.008 |
| /mills\| | | | | |
| lambda | 0.505 | 0.723 | 0.65 | 0.450 |
| rho | 1.000 | | | |
| sigma | 0.505 | | | |

Source: Author's computation, 2019. ***, **, * Significant at 1, 5, and 10 percent significance level, respectively. Note: two-step estimate of rho = 1.979 is being truncated to 1; Number of obs = 277; Selected = 275; Nonselected = 2; Wald chi2 (7) = 34.50; Prob > chi2 = 0.0000.

Equally, marital status is negatively correlated and statistically significant, thus influencing farmers' willingness to pay for IBLI. Married farmers are not willing to pay for IBLI. Their unwillingness can be ascribed to the fact that they consider taking care of their families, their responsibilities, and as a result they are wary of putting resources into a programme that is unfamiliar to them. This result however contradicts the findings of [28]. Their investigation evidenced that marital status is positively correlated and significantly affects the adoption of IBLI at the 5% level.

Similarly, the second stage of the equation (the selection stage) demonstrates that the number of household dependents actively influences the total livestock units a farmer is willing to insure. Household dependency is negatively correlated to the total livestock units (TLUs) a farmer is willing to insure, and this effect was statistically significant. A household head with more dependents is not likely to insure their cattle under the IBLI programme, simply because the head of the family has more dependent responsibilities to take care of, and would rather not divert the resources to buy IBLI at the expense of shirking his responsibility.

Table 9 shows the post estimation and predictive margin of the of heckman selection model, which explains the significance and fitness of the model used.

**Table 9.** Post estimation of Heckman selection model—two-step estimates. Predictive margins. Model VCE: Conventional. Expression: Linear prediction, predict ().

| | Delta Method | Delta Method | | |
|---|---|---|---|---|
| | **Margin** | **Std. Err.** | **z** | **P > \|z\|** |
| constant | 0.103 | 0.032 | 3.16*** | 0.002 |

Source: Author's computation, 2019. ***, **, * Significant at 1, 5, and 10 percent significance level, respectively.

## 5. Conclusions and Policy Implications

The results indicate that a very small proportion of livestock farmers (10.8%) are willing to pay for IBLI at a maximum amount of R600/ TLUs, as the programme is a relatively new concept. The farmer's experience, age, education, marital status and awareness of insurance significantly influence their willingness to pay for IBLI, which could serve as an adaptation strategy and/or agricultural

risk management strategy against climate change variabilities, such as drought and extreme weather conditions. Consequently, the number of household dependents had a negative impact on the TLUs to be insured in the IBLI programme. Thus, awareness of IBLI should be created among livestock farmers as a means of risk management that can offer them a much needed channel to cope with the contrariness associated with the variabilities climate change. Livestock farmers should be educated on IBLI and the benefits of paying the insurance premium. Dissemination of IBLI knowledge should be shared and communicated through the right channels to the livestock farmers. Correspondingly, the policymakers and the stakeholders should consider the introduction of the IBLI concept as an agricultural climate risk policy. Also, government should intervene by offering agricultural grants or subsidies which takes care of or subsidises the amount to be paid for IBLI.

## 6. Declarations

**Ethics approval and consent to participate:** Ethics approval and consent were granted by the university college that deals with ethical issues and clearance.
**Consent for publication:** Consent form has been obtained.
**Availability of data and material:** Data are available upon request.

**Author Contributions:** Conceptualization, O.S.O. and M.A.A.; methodology, O.S.O.; software, O.S.O.; validation, O.S.O., M.A.A. and S.S.T.; formal analysis, O.S.O.; investigation, O.S.O. and M.A.A.; resources, M.A.A.; data curation, M.A.A.; writing—original draft preparation, O.S.O.; writing—review and editing, O.S.O.; M.A.A. and S.S.T.; visualization, O.S.O.; M.A.A. and S.S.T.; supervision, M.A.A.; project administration, M.A.A.; funding acquisition, M.A.A. and S.S.T. All authors have read and agreed to the published version of the manuscript.

**Funding:** The research was funded by the Agricultural Research Council, Climate Change Collaboration Centre. This is a research collaboration entity for UNISA, University of Pretoria and Agricultural Research Council of South Africa.

**Acknowledgments:** The authors acknowledge the anonymous reviewers for improving the quality of this paper. The authors appreciate all the participants who contributed to this research, including ARC, University of Pretoria and University of South Africa.

**Conflicts of Interest:** The authors declare there is no conflict of interest.

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
