# Peer review of "Farmers’ Willingness to Pay for Index-Based Livestock Insurance in the North West of South Africa"

_climate, doi:10.3390/cli8030047_

Round 1
Reviewer 1 Report
Dear Authors,
See the attached comments/suggestions.

Author Response
Response to Reviewer 3 Comments
Point 1: “Smallholder farmers’ willingness to pay for index-based livestock insurance: evidence from cattle farmers in North West, South Africa” The title is long. Please make it short and precise. Like “Farmers’ willingness to pay for Index-Based Livestock Insurance in North West of South Africa.” Or related title (MINOR COMMENT).
Response 1: Changes were made.
Point 2: The abstract is long. Make is short focusing the main points (MINOR COMMENT).
Response 2: Changes were made.
Point 3: Replace Contingency Valuation Method to Contingent Valuation Method (MINOR COMMENT).
Response 3: Changes were made.
Point 4: Farmer’s willingness to pay for index-based insurance at a maximum price of R600. Please rephrase as follows the maximum price R600 (max WTP) who accepted the Index-Based Livestock Insurance (MINOR COMMENT)
Response 4: Changes were made.
Point 5: Please use Rand and USD in Bracket for first time in the Abstract and the body (MINOR COMMENT)
Response 5: Changes were made.
Point 6: Only 10.8% of the famers were willing to pay, which is very low. Please justify this in the result/discussion section (MINOR COMMENT).
Response 6: Changes were made.
Point 7: Page 2, replace ‘Thence’ by ‘Hence’ (MINOR COMMENT).
Response 7: Changes were made.
Point 8: Please provide clearly the gap of the current research that filled. How the current study differ from other studies in the context of livestock insurance (MAJOR COMMENT).
Response 8: Changes was made.
Point 9: In abstract, authors showed that Heckit sample selection model employed in the analysis. In the methods section, Heckman Selection Model (two-steps) model was employed. For consistency and clarity to the reader use ‘Heckman selection Model (two-step) (MINOR COMMENT).
Response 9: Changes were made.
Point 10: Please include the references for: “Likewise, many researchers who have adopted CVM have been able to give a tangible result and predict the number of non-market goods, such as connections to water supply systems, improved conditions, and revenue for the local water authority” (REF) AND for
Response 10: Changes were made.
Point 11: “Heckman’s two-step model is preferred for this study because the estimators from this procedure are consistent and asymptotically normal, moreover, it is more robust than the Heckman MLE approach” (REF) (MINOR COMMENT).
Response 11: Changes were made.
Point 12: In Heckman model (two-step), it is suggested to have at least one variable that affects the selection equation without affecting the outcome equation (Wooldridge, 2002. Econometric analysis of cross section and panel data). Otherwise, the estimates lack accuracy as IMR correlated with explanatory variables. Please address this issue (MAJOR COMMENT).
Response 12: We did include two variables: labour use, and own the land, on the selection equation but they were rarely used or kicked out by STATA during the analysis as they are multicollinear.
Point 13: What are the criteria for choosing the following covariates in the Heckman Model specification (farming experience, age, education, household dependents, marital status, access to credit and awareness to insurance)? Variables such as geographic differences (location) are strongly associated with climate variability that is not accounted in the model. Household income is also another factor that affect the willingness to insurance scheme. Please take into account location, herd size and income in the model and support with references. Authors can compare the new result with the current table and locate the new table result in Appendix (MAJOR COMMENT).
Response 13: farming experience, age, education, household dependents, marital status, access to credit and awareness to insurance, all these were motivated by literature. Changes were made to cite them. However, we did try to include farm income, total livestock unit, in the model but they were omitted on the STATA output results.
Point 14: Please correct the notation ‘1ia’ in the phrase “The Heckman model assumes that the error terms ui and 1i a”’ Please check all notations, editorial issues, references vs citations (MINOR COMMENT).
Response 14: changes were made.
Point 15: Please develop a paragraph about the ‘Design of IBLI’ (what is index insurance, how farmers understand about the index insurance, figures on mortality of livestock in the region, brief about measured Normalized Difference Vegetation Index (NDVI) used to assess green biomass). Authors can bring some information from introduction to this section (MINOR COMMENT)
Response 15: changes are made.
Point 16: Please improve the formatting of the tables and the graph. Except main tables, please locate all other tables and the graph to the Appendix (MINOR COMMENT).
Response 16: I think the journal has a particular style. However, this can be improved if it's ok by the editor.
Point 17: This section is more of result with limited implications. The discussion is shallow with limited references. Please rewrite main implications of the result by supporting additional literatures from relevant studies (MAJOR COMMENT).
Response 17: changes were made by beefing up the argument in the discussion section.
Point 18: Authors include the following sentence: “the policymakers and the stakeholders should consider the introduction of the IBLI concept as agricultural climate risk policies”. However, insurers are commercial that need financial return but most famers are not willing to participate in IBLI (only 10.8% willing to pay). Is the low participation of farmers associated with awareness, inability to pay or protest to the insurance scheme? The policy recommendation is not in line with the reality (low participation is not viable to attract insurance companies). Please justify this in the result/discussion section and brief its implication in the conclusion (MAJOR COMMENT).
Response 18: The willingness to pay is determined by the low participation of farmers associated with awareness, farming experience, age education and marital status of the farmers. These were mentioned in the result/discussion. However, we have added some points to beef up the implications on the conclusion section.
Point 19: There are recent related journals (cattle insurance studies) in Africa/developing countries but most of them are not incorporated in the current study. Please search those updated literatures and discuss in the current study (MAJOR COMMENT).
Response 19: Changes were made.

Reviewer 2 Report
Dear author(s), I really appreciate your study.
I found much interesting this topic and your analysis, actually the index insurance represents a good policy instruments.
I have some suggestions to improve your works.
Introduction: In the introdction section is necessary to improve some aspects. 1) the importance of insurance instruments to coverage the damege produced by climate change.
In this sense see and cite:
Porrini, D., Fusco, G., & Miglietta, P. P. (2019). Post-adversities recovery and profitability: The case of Italian farmers. International journal of environmental research and public health, 16(17), 3189.
2) Add some recently studies that we have used this approach.
In particular for the methodology and conclusion sections:
Matsuda, A., Takahashi, K., & Ikegami, M. (2019). Direct and indirect impact of index-based livestock insurance in Southern Ethiopia. The Geneva Papers on Risk and Insurance-Issues and Practice, 1-22.
John, F., Toth, R., Frank, K., Groeneveld, J., & Müller, B. (2019). Ecological vulnerability through insurance? Potential unintended consequences of livestock drought insurance. Ecological economics, 157, 357-368.
In the methodology section it is not much clear why have you used R600? ( this comments regarding the other thresholds, also) and how is compute the WTP through equation 2 and 3.
Please explain better this concept.
Conclusion section:
Is necessary to improve this section, identify the right economic policy solution.
Author Response
Introduction: In the introduction section is necessary to improve some aspects.
1) the importance of insurance instruments to coverage the damage produced by climate change. Response: I have added the importance and cited the reference given.
2) Add some recently studies that we have used this approach. Response: I did add the recent studies that used this approach. Included in the revised manuscript is 4 research, which I cited.
In the methodology section, it is not much clear why have you used R600? (this comments regarding the other thresholds, also) and how is compute the WTP through equation 2 and 3.
Please explain better this concept. Response: R600 is the maximum amount that a farmer is willing to pay for index-based livestock insurance. This concept of computing WTP in equations 2 and 3 was determined by asking the individual farmer to put the amount, willing to pay for IBLI in the space provided in the questionnaire since it is a single-bounded model with the open-ended contingent valuation. The minimum amount an individual farmer is willing to pay is R400 while R600 is the maximum amount. However, I have explained in the manuscript following your suggestions.
Conclusion section:
Is necessary to improve this section, identify the right economic policy solution. Response: I feel like the right economic policy solution is to create awareness of IBLI since the program is relatively new in South Africa. However, I included in this section according to your suggestion, the economic policy in which I stated that: government should intervein by offering agricultural grants or subsidies which take care or subsidies the amount to be paid on IBLI.

Reviewer 3 Report
As stated in the abstract, the paper investigates farmers’ willingness to pay for index-based livestock insurance and its determinants using the North West province of South Africa as a case study. Cross-sectional data obtained from 277 cattle farmers have been used to assess the farmers' willingness to pay using the Heckman sample selection model.
The paper does fail to adequately present the methodology, the methodology seems not always appropriate and the conclusions are not based on the results of the analysis.
Because of this I suggest rejecting the paper.
Please, find below my comments that may help Authors to improve future analyses.
Problems with the methodology*
The paper does not explain how the total livestock units to insure are considered. Does it refer to the number of livestock heads or the number of farms? While eq (1) is potentially useful, I was not able to find how many cattle farmers are in the region, hence to infer whether a sample of 277 is enough (page 4). How the reliability test has been done (page 4)? Text on page 5 is very similar to FAO: Introduction and General Description of the Method of contingent valuation. Please perform a plagiarism check. In eq (2) some parameters are not defined. A description and reference for the single-bounded model is required. In the table last line of page 5 it is not clear why di = 1 if di* >= 0. In particular, why you impose di=1 when di*=0? On page 6 you refer to employment and wages as e.g. but these are out-context here. What is the value-added of (5) and (6)? If useful, please specify what is Z and provide references. I am not convinced that the OLD for stage 2 is correct provided that the support for the normal distribution goes to – inf on the left, while the WTP is censured to zero. Table 5 repeats the same data as Table 4. Figure 2 seems not useful to me and no explanations are provided. Are these factual data and/or derived from the model? The fitted distribution spread over a too wide range of values. In my opinion, two important explanatory variables are missing: relative farm size and presence of off-farm incomes. Both affect the risk attitude of the farmers. Tables 6 and 7 should be better designed and explained. As such these are not clear at all. Results in Table 8 under TLU willing to insure should be explained. I am very worried that the first stage of the model is very weak. This should be clarified to avoid that the whole model is not robust enough: in particular, the coefficient reported as lambda is not significant. Isn’t this a problem? This may explain why results reported as marginal effects after Heckman are exactly the same as those reported at the beginning of Table 8. Please, note that the significance of the coefficients is not reported correctly. It is not explained why the Delta method is used and why only the constant is reported. Finally, it is not clear what is the meaning of Figure 3 and why a different type of plot has not been used.
Conclusions:
Why do you refer to 10.8% of livestock farmers are willing to pay a maximum amount of R600/TLU? According to table 4, these are only 2.9%. Why should we increase the use of the IBLI? Of course, this can be useful for the Institution selling the IBLI, but is it useful for farmers? Cannot be simply that the low WTP suggests this is not a useful tool for farmers? How the results of the analysis support your policy implication?Author Response
Response to Reviewer 2 Comments
Point 1: Why do you refer to 10.8% of livestock farmers are willing to pay a maximum amount of R600/TLU? According to table 4, these are only 2.9%. Why should we increase the use of the IBLI? Of course, this can be useful for the Institution selling the IBLI, but is it useful for farmers?
Response 1: We didn’t increase the use of IBLI, what we meant here was that the maximum amount to pay for IBLI was R600/TLU. On the contrary, R400 is the minimum amount some farmers are willing to pay for IBLI, some R500 but the maximum amount as specified in table 4, was R600/TLU.
Point 2: Cannot be simply that the low WTP suggests this is not a useful tool for farmers? How the results of the analysis support your policy implication?
Response 2: This does not necessarily suggest that low WTP is not useful. This is the reason we are trying to find out the determinants. Also, we should bear in mind that, the program is relatively new. It has been practiced in some parts of Africa, like Ethiopia but not in South Africa at the moment. Farmers do pay for traditional insurance which we already emphasized the disadvantages and why people are not buying it. Hence, we want to find out how many farmers would be willing to buy and for how much. However, the results say it all, that awareness needs to be made, proper dissemination of this program will also help. In my opinion, I believe the introduction and acceptance of this program will help both the policymakers and the shareholders including the farmers to fight the threat posed by climate-related events, and enhance food security.
Point 3: Figure 2 seems not useful to me and no explanations are provided. Are these factual data and/or derived from the model?
Response 3: The graph explains the willingness to pay for IBLI. It’s just a histogram buttressing Table 5. However, we can add a little explanation of what the histogram is explaining. Battershill, if you want us to remove the graph, that we can do as well.
Point 4: The paper does not explain how the total livestock units to insure are considered. Does it refer to the number of livestock heads or the number of farms?
Response 4: Total Livestock Units are livestock numbers converted to a common unit (in 2005). Conversion factors are: cattle = 0.7, sheep = 0.1, goats = 0.1, pigs = 0.2, chicken = 0.01. This can be calculated manually, however, for the purpose of this research, we have a spreadsheet program that actually calculates the TLU by punching or inserting the livestock numbers. Thus, it automatically generates the TLU.
Point 5: I was not able to find how many cattle farmers are in the region, hence to infer whether a sample of 277 is enough (page 4).
Response 5: The accurate data of the numbers of livestock in the region was not given, but we were able to get the accurate number of livestock each sampled farmers possessed. This was calculated to give us the TLU used in the model. Similarly, I have included in the manuscript the population size, sampling procedure on how we determine the sample size (sampled farmers and their number of livestock).
Point 6: How the reliability test has been done (page 4)?
Response 6: This was conducted by administering to the farmers to check how dependable or consistent the questionnaire is. The farmers were used to pre-test the questionnaire. However, these farmers are not part of the sample size used in the study.
Point 7: Text on page 5 is very similar to FAO: Introduction and General Description of the Method of contingent valuation. Please perform a plagiarism check.
Response 7: This was done during the submission of the manuscript. However, we would perform the plagiarism check again and rewrite it.
Point 8: A description and reference for the single-bounded model are required.
Response 8: This has been provided in the revised version.
Point 9: In the table last line of page 5 it is not clear why di = 1 if di* >= 0. In particular, why you impose di=1 when di*=0?
Response 9: This is because stage 1 is a binary response. The single bounded model as explained, is binary in nature (WTP = 1 or not WTP = 0).
Point 10: On page 6 you refer to employment and wages as e.g. but these are out-context here.
Response 10: Changes have been made in the revised manuscript.
Point 11: Tables 6 and 7 should be better designed and explained. As such these are not clear at all.
Response 11: This shows an association between variables (continuous and dummy variables) in relation to the willingness to pay for IBLI. We included this, following Gebrekidan et al. (2019), who conducted research on index-based insurance. The citation was included in the revised manuscript.
Point 12: Table 5 repeats the same data as Table 4.
Response 12: This is not exactly repetition, as Table 4 shows the amount willing to pay for IBLI, while Table 5 shows how many, percentage of farmers willing to pay for IBLI. There is a difference between the two Tables.
Point 13: In eq (2) some parameters are not defined.
Response 13: This shows the equation for the willingness to pay for a commodity. In my opinion, I don’t think something is missing. This citation is provided in the revised manuscript.
Point 14: What is the value-added of (5) and (6)? If useful, please specify what is Z and provide references.
Response 14: The two-equation complement each other. Reference was provided in the revised manuscript.
Point 15: I am not convinced that the OLD for stage 2 is correct provided that the support for the normal distribution goes to – inf on the left, while the WTP is censured to zero.
Response 15: The stage 2 model is a continuous variable of total livestock units to insure. This was regressed against the independent variables, in Table 8 provided.
Point 16: Results in Table 8 under TLU willing to insure should be explained. I am very worried that the first stage of the model is very weak. This should be clarified to avoid that the whole model is not robust enough: in particular, the coefficient reported as lambda is not significant.
Response 16: The results were explained on page 9. For the first stage, it was a binary outcome. Any suggestions will be appreciated for us to do. However, in regards to the lambda being insignificant, from my understanding, the presence of a selection effect is tested by the significance of the inverse Mills ratio; which Stata provides when the two-step option is specified. The significance of lambda alone does not indicate sample selection bias but also suggests that the error terms in the selection and primary equations are correlated. In our study, not significant could be that we don’t have enough data to detect it, as this is obvious in Table 8. It does not necessarily mean that the model is not good or fit enough.
Point 17: Isn’t this a problem? This may explain why results reported as marginal effects after Heckman are exactly the same as those reported at the beginning of Table 8. Please, note that the significance of the coefficients is not reported correctly. It is not explained why the Delta method is used and why only the constant is reported.
Response 17: I don’t think this is a problem as we are looking at the value of X in Table 8. The delta method is a change, which is generated by Stata using a command.
Point 18: The fitted distribution spread over a too wide range of values. In my opinion, two important explanatory variables are missing: relative farm size and presence of off-farm incomes. Both affect the risk attitude of the farmers.
Response 18: I do agree with you. Farm size was added in the model but it creates multicollinearity, which was rarely used while running the analysis with Stata. In other words, it was removed.

Round 2
Reviewer 1 Report
The paper is interesting for readers of climate. Authors addressed some of my concern. However, there are still major comments that is not addressed. I have 'Noted' following the previous comment. I hope this helps!

Author Response
Response to Reviewer 1 Comments
Round 2
Question: In Heckman model (two-step), it is suggested to have at least one variable that affects the selection equation without affecting the outcome equation (Wooldridge, 2002. Econometric analysis of cross section and panel data). Otherwise, the estimates lack accuracy as IMR correlated with explanatory variables. Please address this issue. Response: Changes have been effected and run the model analysis.
Question: Design of Index-Based Livestock Insurance (IBLI): Response: This has been added.
Figures on mortality of livestock in the region: Response: The study provides no exact figures on mortality rate as a result of climate-related events. Since the study has not implemented NDVI. The index-based is relatively new and yet to be established, hence, the study seeks to explore the willingness of farmers to pay for IBLI after explaining the concept and how many livestock units would they be willing to be insured.
Question: Please improve the formatting of the tables and the graph. Except main tables, please locate all other tables and the graph to the Appendix. Response: Changes have been effected.
Question: “it is not clear what is the meaning of Figure 3 and why a different type of plot has not been used.” Please provide a point-to-point response to the comments when submitting the revised manuscript. Response: Figure 3 has been removed.
Question: Reference list to be checked: Response: I have made changes and updated the reference list.
Question: NOTE: The implication is still insufficient in the discussion. Moreover, only 10.8% of the farmers are willing to pay is a major concern that is not addressed. Response: Am not sure what you would need me to do here as regards this. The IBLI is not yet implemented in South Africa. The study only tries to explore the factors that can influence the IBLI should it be introduced as indicated in the manuscript. Such identified factors will help the policymakers and extension officers when implementing or introducing IBLI. I don’t except the willingness to be more since it is a relatively new concept which the farmers have not tried or heard about, and this is stated in the manuscript.
Question: This section is more of result with limited implications. The discussion is shallow with limited references. Please rewrite main implications of the result by supporting additional literatures from relevant studies. Response: Some authors and research articles suggested by the previous reviewers have been added. I also improved the result section in my previous revision submission.
Question: There are recent related journals (cattle insurance studies) in Africa/developing countries but most of them are not incorporated in the current study. Please search those updated literatures and discuss in the current study. Response: To the best of my knowledge, I have included them in the revision submitted.

Reviewer 2 Report
Dear author(s) well done!
please check if all references are in the references list!
I really appreciate your corrections.
Author Response
I have made changes and updated the reference list. Thank you.
This manuscript is a resubmission of an earlier submission. The following is a list of the peer review reports and author responses from that submission.